# Research of Machine Learning Algorithms for the Development of Intrusion Detection Systems in 5G Mobile Networks and Beyond

**DOI:** 10.3390/s22249957

**Published:** 2022-12-17

**Authors:** Azamat Imanbayev, Sakhybay Tynymbayev, Roman Odarchenko, Sergiy Gnatyuk, Rat Berdibayev, Alimzhan Baikenov, Nargiz Kaniyeva

**Affiliations:** 1Faculty of Information Technology, Al-Farabi Kazakh National University, Almaty 050040, Kazakhstan; 2School of Information Technology and Engineering, Kazakh-British Technical University, Almaty 050000, Kazakhstan; 3Information Security Laboratory, Almaty University of Power Engineering and Telecommunications, Almaty 050013, Kazakhstan; 4Department of Telecommunication and Radioelectronic Systems, National Aviation University, 03058 Kyiv, Ukraine; odarchenko.r.s@ukr.net (R.O.);

**Keywords:** intrusion detection system, 5G, dataset, machine learning, deep learning, cybersecurity

## Abstract

The introduction of fifth generation mobile networks is underway all over the world which makes many people think about the security of the network from any hacking. Over the past few years, researchers from around the world have raised this issue intensively as new technologies seek to integrate into many areas of business and human infrastructure. This paper proposes to implement an IDS (Intrusion Detection System) machine learning approach into the 5G core architecture to serve as part of the security architecture. This paper gives a brief overview of intrusion detection datasets and compares machine learning and deep learning algorithms for intrusion detection. The models are built on the basis of two network data CICIDS2017 and CSE-CIC-IDS-2018. After testing, the ML and DL models are compared to find the best fit with a high level of accuracy. Gradient Boost emerged as the top method when we compared the best results based on metrics, displaying 99.3% for a secure dataset and 96.4% for attacks on the test set.

## 1. Introduction

Confidential information is often stored, transmitted, and processed in global networks. In this regard, the security of network systems is becoming increasingly important. The new fifth generation (5G) network architecture offers densely distributed storage, computing, and networking capabilities that provide more versatile services than previous generations, supporting a wider range of cases and applications. The scope of the network is being expanded by the increasing number of users (mobile devices, Internet of things), and it is expected that users will utilize 13 times more data in 2025 than now. This is forecasted based on an expected 21 billion devices connected to the Internet in 2025, in comparison to 7 billion [1] devices connected to the network today, which also implies an increase in hacker activity in the future. IoT is applied in various fields of research and is used in various applications such as Healthcare, Smart Grid, Transportation, Smart Home and Building, Smart Cities, Agriculture, Industry Automation, and the Military [2]. After 2030, wireless applications will require much higher data rates (up to 1 Tbps), extremely low end-to-end latency (<1 ms), and extremely high end-to-end reliability (99.99999%) [3]. However, such a great digital evolution is only possible with the next generation of 5G and 6G mobile networks. The migration to the cloud, virtualization, and the majority of network functions becoming software-assisted in 5G wireless networks and beyond is increasing the security risk of accessing core networks [4]. In the same way, the rapid development of technology creates new security issues. As the main architectural pillar of 5G networks is to create programmable and configurable network components using software, it means one has to carefully check the software code before deployment in order to protect resources and user data. Hackers can exploit open vulnerabilities in the supply chain, which can lead to serious attacks which affect the entire network infrastructure. In the 5G network architecture and beyond, there has been a paradigm shift from the concept of dedicated network resources for dedicated network functions to the more dynamic virtualization, cloud organization, and orchestration of software-defined network functions from network resources [5]. All of these factors have the effect of increasing network insecurity and user data risk if malicious attacks are not detected in real time. In this context, machine learning and deep learning [6,7] are expected to play a vital role in the use of automated intelligence in 5G and other wireless networks. While 5G is well known for its cloud-based, microservices-based architecture, the next generation of the network, 6G is closely associated with intelligent orchestration and network management. Hence, the role of artificial intelligence (AI) in the 6G paradigm is of prime importance. AI is key to next generation 6G mobile networks, and ensuring its security is critical to realising the 6G concept. AI-enabled 5G and 6G mobile network security provides intelligent and reliable security solutions [8]. Nowadays, the concept of intelligent analytics needs to be implemented in all types of wireless networks, from local area networks to remote clouds. Network traffic prediction and estimation is a necessary part of network operations and management, such as congestion control, routing, resource allocation, and service level agreement management, as well as many other network responsibilities and functions [9]. Due to this chain of events, strong and effective security measures are required to create a safe and secure environment for users, but it is currently difficult to prevent attacks through passive security policies, firewalls, or other mechanisms. In addition, with the introduction of 5G technology, there will be security risks for older generations of mobile devices. Therefore, along with traditional security tools, such as firewalls, intrusion detection systems (IDS) are becoming increasingly important to help protect systems proactively. It is known that an IDS can collect traffic data (i.e., activity) and can analyse the received information. The intrusion detection system will be based on anomalies. This means that the system must go through a "familiarisation" period, during which it learns and remembers the current state of the infrastructure. What it learns becomes the benchmark against which the system will be guided in the future. In our case, we will train the system based on datasets. When monitoring a network, data is gathered from network packets [10,11,12]. Network attacks are carried either by embedding malicious code or analyzing network packets to gain information. Attacks can take place on either the server that processes all network transactions or on the system node that actually performs network activities. Actions can also be taken to exploit weaknesses in the system. Technologies such as machine learning and deep learning lead to improved intrusion detection systems (IDS) [13,14].

In this regard, in recent years researchers have explored the possibility of using artificial intelligence (AI) techniques to develop efficient IDS applications. In fact, machine learning methods have become one of the most promising tools for studying a wide range of complex issues, given the rapid growth of network traffic and security risks [14,15,16,17]. Research of the use of artificial intelligence technologies, in particular machine learning and deep learning, in network intrusion detection systems (NIDS), is a relevant topic but is still in its infancy, and there is still great scope for exploring these technologies in NIDS systems to effectively detect network intruders.

### 1.1. Background Analysis

Network security is one of the most discussed and important issues in a rapidly evolving society, as it affects the interests of many stakeholders. The rapid evolution of 5G mobile networks creates new risks, threats, and vulnerabilities in the system of which attackers can take advantage [18,19]. The ENISA report examined the challenges, vulnerabilities, and attacks on 5G networks [20], and the transition to 5G will involve several phases, according to the 3GPP roadmap. One of these phases, 5G Non-Standalone, combines the use of the new 5G radio and LTE core network. As a result, these networks will inherit all the vulnerabilities of LTE networks. Studies show that LTE networks are vulnerable to denial of service (DoS) [21,22]. The best practice to defend against these attacks is to use virtual network security tools such as antivirus, virtual firewalls, or IDS/IPS to achieve a level of security comparable to traditional networks. In addition, machine learning (ML) enabled solutions can be used to detect attack traffic (e.g., DoS attacks) and distinguish it from normal traffic so that it can be handled accordingly [23]. Researchers and developers have a lot of work to do to ensure robust end-to-end security. Artificial intelligence (AI) and machine learning (ML) can play a vital role in the development and automation of next-generation mobile networks. The main advantage of the 5G network is its high data transfer rate, and it is more effective to use AI and machine learning to prevent and detect a wide range of threats from different points [19]. Fifth generation technologies, including Multi-access Edge Computing (MEC), SDN, NFV, and network slicing, are still relevant to 6G networks. Therefore, their associated security matters remain. For example, the most severe security concerns related to SDN include vulnerabilities on the SDN controller, interfaces, and SDN applications platforms. Security obstacles associated with NFV include attacks on virtual machines, hypervisors, and virtual network function (VNF) managers. Finally, MEC is vulnerable to physical risks, DDoS, and the enormously distributed structure of 6G systems [24]. The place of artificial intelligence in the 6G network architecture should also be taken into consideration. After all, artificial intelligence will appear in all parts of the network, including the borders of cells and, possibly, user devices. Under these conditions, the possibility of using these algorithms in the tasks of detecting and preventing cyber attacks becomes an obvious advantage.

Researchers have studied technologies, scenarios, and applications using artificial intelligence to secure 5G wireless networks. They have also come to the conclusion that AI can significantly increase the security of a distributed ad hoc configuration of the network infrastructure that provides various network functions. However, more thorough research is required before AI fully takes over the digital automation of mobile networks [25].

Only artificial intelligence (AI) tools, especially machine learning (ML) and deep learning (DL) [26,27,28,29], can handle the real-time analysis of the huge volumes of data traffic that is generated in fifth generation networks. Despite the great possibilities of creating self-managed networks with the help of AI, attacks on algorithms can lead to significant performance degradation and network failures [30].

### 1.2. Related Works

This paper explores the potential of machine learning in IDS to secure 5G networks. It is anticipated that AI will be a key enabler of 5G and other networks [31]. In the past it has become obvious that artificial intelligence (AI)-based techniques play a prominent role in the ensemble development for intrusion detection and have many benefits over other techniques [32]. Here, an updated general review of ensembles and their taxonomies has been presented. The paper also presents the updated review of various AI-based ensembles for IDS (in particular) during the last decade. Various IDS systems exist currently and the authors have presented an in-depth review of intrusion detection systems (IDS) for the IoT from 2015 to 2019 [4]. In [33], various AI based techniques have been reviewed focusing on the development of IDS. A lot of studies were devoted to the investigation and comprehensive analysis of different approaches for detecting different attacks in different conditions depending on the available data collected [34,35,36,37]. A framework to build and operate AI-based intrusion detection for in-situ monitoring was described and analysed in detail in [38].

A large number of papers analyse different IDS datasets and one study provides information on the latest CIC IDS 2017 dataset. The publication discusses the application of IDS as well as potential future research directions [14]. In the same way, the researchers conduct experiments with two reference datasets, namely NSL-KDD and CICIDS2017 [6]. Another article discusses CIC-IDS-2017 and CSE-CIC-IDS-2018, as well as a review of the ML and DM algorithms used for IDS. These are the most recent datasets which provide characteristics of network attacks, which include new types of attacks [7]. Deep learning is already widely used to solve the detection problems of various network attacks [39,40,41]. In [42], the Deep Neural Network on NSL-KDD dataset was researched for effective attack detection. 

The performance of IDS is one of the key factors. The researchers [43] in this study focused on improving the performance of DNN-based IDS by providing a unique feature selection method that combines statistical significance using standard deviation and the difference of mean and median. The effectiveness of the proposed approach is evaluated using three intrusion detection datasets: NSL-KDD, UNSW NB-15, and CIC-IDS-2017.

In addition, the criteria that are set for the datasets play an important role. Markus Ring [44], for example, discusses common aspects of dataset descriptions and divides them into five categories. This study provides a focused assessment of datasets for network-based intrusion detection, as well as specifics on the underlying packet- and flow-based network data. The report identifies 15 alternative parameters for assessing individual dataset applicability for specific evaluation scenarios.

Furthermore, it was shown that one of the best solutions for monitoring and detecting threats in 5G networks is to use AI-based IDS trained on big data [45]. Similarly, the main security problem is the development of a methodology for detecting malicious activity, due to the fact that it is necessary to update the database with malicious traffic for AI training [46]. In the literature, many researchers have proposed different ensembles by considering different combination methods, training datasets, base classifiers, and many other factors [47,48]. However, the task to identify the most correct usage of datasets remains open. Researchers have proposed a methodology for integrating intrusion detection systems into the standard 5G architecture [49].

However, no cases on network security have shown applications of various deep learning algorithms in real-time services beyond experimental conditions in 5G networks.

We propose a DNN-based intrusion detection and classification system, which takes into account statistical indicators to evaluate the performance of models. After pre-processing the data, we performed feature engineering to select and transform features that can be used to build the model. Two public datasets are used for implementation, such as CICIDS2017 and CSE-CIC-IDS2018.

### 1.3. Problem Statement

The architecture of traditional networks has not changed for decades, which brought many problems and highlighted several security issues, and the development of 5G networks has raised even more concerns about the future structure of the Internet.

To advance with this new technology, the use of software-defined networks is essential to begin the successful deployment and implementation of a powerful wireless world [50]. Several researchers have repeatedly said that the SDN architecture has many advantages as it provides many solutions to the problems of legacy network infrastructure, which has attracted the attention and interest of scientists [51,52]. Thus, it is seen as a new software-based network architecture that could offer significant benefits for 5G networks. The most notable features of this network are that it is low cost, flexible, expandable, and it increases the size of its infrastructure without the complexity of a traditional network. This architecture consists of three main layers (control plane, controller, and application plane), and all operations in this architecture are controlled by the controller [53]. Since this element is considered to be the brain of the network, it is completely isolated from the network and if attackers attack it, it will lead to the downfall of the entire network. Accordingly, the controller is the most malicious part and the most vulnerable to attacks. In response to this threat, the need to develop an intrusion detection system (IDS) has emerged and grown. This is because it constantly monitors the network and creates a traffic pattern that enables it to detect behavior or traffic patterns that deviate from the normal pattern [54]. 

In this paper, we offer deep learning technologies as they can quickly and accurately identify a wide range of attacks. However, conventional types of Machine Learning (ML) algorithms and many types of Deep Learning (DL) algorithms are initially used to evaluate them based on various criteria (Accuracy, F-score, Recall, Precision, etc.) [55,56]. To train our classifiers based on strongly related features, we used feature selection approaches. This data can then serve as a basis for new researchers willing to start exploring this promising area. The main contributions of this article are listed below:Overview of available datasets for building smart LEDs. A comparative analysis between them is also presented, highlighting their metadata, types of attacks, format, etc.An overview is provided on the work of similar topics on the application of ML and DL in NIDS, which have previously been explored by other researchers.Comparison of the performance of Logistic Regression (LR), Gradient Boosting, Random Forest (RF), Autoencoder, and Deep Neural Network (DNN) with hyperparameter search.Unresolved issues in the development of NIDS based on machine and deep learning are highlighted.

## 2. Proposed Methodology and Model Classifiers

Implementing an intelligent system to detect intrusions into the core network can be achieved through software-defined security. This is because the two main components of a 5G network, RAN and the core network, are virtualized and are fully software-defined. Therefore, it is possible to go in one direction and create an automated security system. The idea is that copies of the traffic from the backhaul connection and the core network are sent to SDS for analysis [57]. It should be noted that the copies of the traffic do not affect performance in any way, while the network is being analysed. However, before it can be determined whether or not the traffic is anomalous, the data must be pre-processed to make it more readable and easier to use for machine learning or deep learning models. The anomalies are then analysed with the appropriate algorithms and the results are sent to the Policy Manager database. The results are then forwarded to the VNF Manager, which updates the module IDS. Here, the model processing time plays the most important role in presenting the final results. In other words, this helps determine when the template needs to be run so that the module policies are up to date. Thanks to this technique, it is possible to automate the detection, the update of the attack database, and the actions taken to defend the network against intruders. 

Figure 1 shows a possible implementation of the IDS module in the 5th generation mobile network.

One of the most important factors in the development of ML-DL-based IDS in SDN is the appropriate selection of datasets. There is an obvious lack of studies on datasets used in ML-based IDS-SDN research. Nevertheless, relatively few of them have applicable types of attacks and properties that could help in implementing models in practice. In this part of the section, we address the main difficulties that researchers encounter in developing an intelligent intrusion detection system.

First of all, it is very difficult to collect reliable data for research. This is because technology changes several times a year, which increases security threats and updates the list of new attacks. As a result, datasets quickly lose their importance and value in the cybersecurity society. The second problem is the integrity of the dataset. This means that researchers need to include not only CSV files, but also audit logs and raw data from the network. Audit logs can be used to find important information about cyber-attacks, and raw data improves threat detection. The next point to consider is the types of attacks. As technology advances, new types of attacks emerge as hackers adapt their attacks to current systems or software, creating a vicious cycle. In this case, two methods come into question: the use of new datasets or the dataset generator, which adapts like a hacker and creates corresponding attacks.

In addition, the generated datasets must be as realistic as possible if stakeholders are to use the model in production. In other words, they must contain normal traffic from various end-user workstations and servers. Otherwise, the trained model may not be suitable for a particular network. It should be noted that data privacy is also considered in the dataset. Although the most trusted data sources are the providers’ mobile networks, they are not always willing to share their data (audit log or network logs) as this violates privacy policies. Therefore, researchers do not train and test their model with real network traffic data, but usually use popular datasets where the data is modeled [6,14,40,42].

The need for labeling is also high [58]. This is true whether it is supervised or unsupervised learning, as labeling is required to calculate the accuracy of the algorithm used. In fact, experts use cyberspace to collect secure network activity before using the data to attack network traffic. Therefore, they set up normal traffic first and then attack. Some experts insert attacks into normal traffic, while others do a manual tagging, which makes the latter process more laborious. Finally, the dataset needs to be widely accepted by the research community in order for the scientific work to be appreciated. Without this support, the dataset can only be used in a few research projects.

As written in the related works, modern scientists have done a lot of research on dataset analysis for IDS. The algorithms used for IDS are implemented on the DARPA dataset [59], KDD CUP 99 [60], NSL-KDD [47], or UNSW-NB15 [61] in which the network instances are grouped as training and test sets. The CIC-IDS-2017 and CSE-CIC-IDS2018 datasets present a new spectrum of generated attacks based on real network traffic characteristics. Table A1 in Appendix A gives details.

5G networks have become the backbone of the Internet of the future. At the same time, it is obvious that the functioning of this type of network will work according to the principle of the all-over-IP architecture. Under these conditions, it is clear that in new networks, along with a large number of new cyber attacks, there will remain no fewer well-known and quite advanced criminal network attacks aimed at vulnerabilities in network architectures using the TCP/IP protocol stack. Therefore, if we look at the 5G network from an Internet-based perspective, the set of parameters included in the CICIDS2017 and CSE-CIC-IDS2018 datasets can be used to train AI-based IPS/IDS to detect common network attacks. Moreover, many researchers [62,63,64] have considered the prospects of 5G and network architecture based on the SDN principle.

The authors in [7,17] present an exhaustive survey on IDS based on CICIDS2017 and CICIDS-2018 datasets. They examined numerous research papers and compared their performances based on their ML models, computing environments, and several performance parameter scores such as accuracy, precision, recall, the area under the curve, etc. The CICIDS2017 and CSE-CIC-IDS-2018 datasets can be a convincing dataset to evaluate ML-based IDS in the 5G network.

Datasets such as CICIDS2017 [65] and CSE-CIC-IDS2018 [66] have been considered in this study. In the first case, the authors studied the model using the dataset, and in the second case, they evaluated the performance of the model. There are many other datasets available on the Internet to monitor network traffic, but some of them are outdated, inflexible, and have duplicate credentials. Figure 2 describes the features of the two selected datasets. 

Complete form of the CICIDS2017 dataset that contains 3,119,345 instances and 83 attributes containing 15 class labels (1 normal + 14 attack labels). The prevalence of the majority class (Benign) is 83.34% and that of the minority class is 0.00039% (Heartbleed). CSE-CIC-IDS2018 was generated from a significantly bigger network of simulated client-targets and attack machines [7], yielding a dataset of 16,233,002 instances acquired from 10 days of network activity. Approximately 17% of the occurrences are assault traffic.

### Evaluation Metrics

To evaluate the developed IDS models for the SDN based 5G network and then compare them, indicators such as accuracy, precision, recall, and F1-score are used [68].

All these metrics are determined by the following characteristics:True Positive (TP): The malicious flow is classified as ‘malicious’ by the model and the result is a true positive.False Positive (FP): The malicious flow is classified as ‘benign’ by the model, the result is a false positive.True Negative (TN): The benign flow is classified as ‘benign ‘ by the model and the result is True Negative.False Negative (FN): The benign flow is classified as ‘malicious’ by the model, resulting in a false negative result.
(1)$$Accuracy=\frac{TP+TN}{TP+TN+FN+FP}$$
(2)$$Precision=\frac{TP}{TP+FP}$$
(3)$$\;\;\;\;\;\;Recall\;\left(DT\right)=\frac{TP}{TP+FN}$$
(4)$$F1-score=\frac{2\times Recall\times Precision}{Recall+Precision}$$

## 3. Experiments and Result

In this section, we show the machine and the deep learning algorithms they used for their work.

### 3.1. Data cleaning

To ensure that their data is prepared for the analysis phase, the company will benefit greatly from data cleansing, which improves data quality [69]. The process of preparing the data for analysis before designing the model should be done by filtering out unnecessary or misleading information (e.g., data cleaning). Datasets are usually collected and merged into smaller files, which could lead to some duplicates and unwanted items. It is worth noting that an incorrect data collection method can lead to the misrepresentation of data and a decrease in the accuracy of models. In addition, models trained on the wrong datasets may perceive the noise as valuable information, and when it comes to training, it will show a good result. However, when cleaned datasets are input into it, unsuccessful results are displayed.

Consequently, the following manipulations were carried out on the CSE-CIC-IDS2018 dataset:Removing invalid linesRemoving invalid valuesCleanup script

In CSE-CIC-IDS2018, the dataset stores ten separate CSV files, each containing recorded network traffic for one day of operation, named after the day the traffic was recorded. Therefore, one file (i.e., ‘Thursday-01-03-2018_TrafficForML_CICFlowMeter.csv’) is loaded and analysed for the initial analysis of the dataset.

As for the first step, when querying information on columns, the first problem encountered is that pandas outputs all columns as object columns, not numeric columns, which is fine for the most of them. To understand the reason why columns are interpreted as objects, the sub-columns are analysed to reveal individual values (Figure 3). 

A distinct value indicates that the column name exists as a value in the dataset. A visual inspection of the input file reveals multiple occurrences of the title in the file combined with the original data path. This indicates that the file was created by merging multiple CSV files, with repeating titles. To resolve this issue, all header columns are removed from the data frame.

Closer visual inspection of the file reveals the presence of the infinity chain in several rows of this column. The pandas read_csv() method cannot correctly parse this value because it only recognizes inf/-inf strings as a valid representation of infinity. To solve this problem, all occurrences of infinity are replaced by the string ‘inf’.

After correcting the data in one file, the same must be done in the remaining nine files. Therefore, a script was written so that all datasets go through data cleaning:Deleting duplicate headers entered as dataset rows.Replacing occurrences of ‘Infinity’ with ‘inf’.Renaming columns to remove spaces and characters without words.

The script (Appendix B) processes all files in the dataset and saves the output file with a name that describes the type of streaming attack in the file.

### 3.2. Exploratory Data Analysis

The next step is exploratory data analysis (EDA) [70,71,72]. This approach is useful for visualizing data and finding answers to a specific task. 

In this work, the authors discovered the amount of safe and malicious network flows that the dataset contains and the amount of network streams that each type of attack contains. A strong correlation between certain features was examined to understand which features are worth paying attention to.

After conducting an exploratory analysis of the data, the following properties were revealed:The dataset is not balanced, so safe network traffic far outweighs malicious traffic (Figure 4).Another problem with datasets is the types of attacks, some of which are poorly specified (Figure 5). Therefore, it is difficult to recognize these attacks when training multi-class classifiers.When a correlation was established between features in the dataset, a strong correlation was found, suggesting the idea that the dataset contains redundant data. This must be taken into account when selecting and extracting features.Several features were then analysed to find predictors for the binary classifications. Due to numerous features, the visual identification of potential features is not possible. In order to identify features with high predictability, it is proposed to use a functional selection and extraction process such as PCA.

In general, the main predictors for binary classification were found:fwd_seg_size_minbwd_pkts_sack_flag_cntfwd_seg_size_minbwd_pkt_len_min

### 3.3. Building ML Model Prototype

A formal branch of research called "machine learning" focuses on the theory, effectiveness, and characteristics of learning algorithms and systems. It is a highly interdisciplinary field based on concepts from numerous disciplines of science, engineering, and mathematics, including artificial intelligence, optimization theory, information theory, statistics, cognitive science, optimum control, and many others [73,74,75,76]. Machine learning has nearly all scientific fields covered thanks to a wide range of applications, which has had a significant impact on both research and society [77]. Numerous issues, including those related to recommendation engines, recognition systems, computer science and data mining, and autonomous control systems, have been resolved using it [78].

Since 5G generates more data at a faster rate than previous generations, telcos must be able to collect and analyse it at scale. By investing resources in the development of standards and a seamless framework that enables data governance, data integration, a modern data architecture that allows data to be accessed regardless of where it resides, and the ability to perform analytics on the database at any scale, current and future needs for 5G analytics can be met.

### 3.4. Experiment—1

This step used various types of existing algorithms to create binary classifications [79,80] that can distinguish between secure network traffic and malicious traffic based on the CIC-IDS-2018 dataset, for example:Logistic regression [81]Random forest [82]Gradient Boosting [83]

Timestamp and dst_port features were not included in the model, so the attack can be recognized regardless of the time and port of the target performing the attack. To do this, both features were removed from the dataset. After detecting the high correlation, the next step was to delete these properties. To confirm these characteristics, hierarchical clustering was performed on the Spearman rank-order correlations [84]. After choosing a threshold, an attribute from each cluster was stored in the dataset. In the end, the remaining number of features was 31. The reason for removing these features was that they did not affect the predictability of the model in any way, but rather caused noise (Figure 6 shows a correlation heatmap after removing highly correlated features).

After that, there is a step of dividing the dataset into training, evaluation, and testing with ratios of 0.8, 0.1, and 0.1, respectively. Then, it turns out that the dataset is highly unbalanced: class 0–benign makes up about 83% of all samples. For this reason, two metrics were used to evaluate the classifier, namely Recall and Precision.

Since the goal of the classifier is to identify as many attacks as possible, the first indicator was used as the main metric. The second one (i.e., Precision) was used as a secondary classifier, since the number of false positives should be kept to a minimum. This metric must have a value above the 0.95 threshold in order to have a maximum of 5% false positives.

When developing a machine learning model for any project, it is best to start with a baseline model. It is a Dummy model [85] that consistently predicts the most commonly encountered class. In the IDS method, the base model defines the system’s normal or expected actions and compares all network actions or traffic to this base model.

The first experiment was the application of logistic regression algorithm, which are used to observe discrete classes. To use logistic regression, the predictors are scaled using the standard scaler. Although this is a popular algorithm, it has the disadvantage of being sensitive to outliers. In the end, the algorithm surpassed the baseline, showing a weighted recall of 0.88 and a precision of 0.87, which is not enough for practical application. The next algorithm to be evaluated is the random forest classifier implementation from scikit-learn. The Random Forest classifier performs very well with a recall of 0.99 and a precision of 0.99. This group classification (Random Forest) works better than other traditional classifiers to effectively classify attacks even with default values. In the latest algorithm, gradient boosting using the CatBoost library [86] was used. This algorithm also makes predictions based on an ensemble of other algorithms. Its main difference with a random forest is the sequence of tree creation, while in the previous algorithm a decision tree was created for each sample. In the research, we applied a grid search using cross-validation on various hyperparameters performed to determine the optimal parameters (Table 1).

Table 2 shows the evaluation metrics of four algorithms to select a single model and continue the experiment. From the table, the Gradient Boost algorithm slightly outperformed Random Forest, so it passes to the next evaluation stage, namely model testing. The final score shows very good performance, with a recall of 0.99 and a precision of 0.99.

However, Figure 7 shows the misclassifications in the test dataset, which demonstrates that "Infiltration" attacks are often misclassified. Furthermore, minority attack classes are often misclassified. To improve performance, synthetic minority resampling can be applied to the training dataset for these classes.

To ensure that the estimator has the same good performance as shown in the test dataset, additional tests were performed on the CIC-IDS-2017 dataset, which contains the same attack scenarios but is recorded in a different network environment. However, the estimator performed very poorly with data recorded in a different network environment (Figure 8), showing a recall of 0.82 and a precision of 0.80. Moreover, the estimator had an attack recall of only 0.26, which is not sufficient for real-world networks. This result suggests that data from one network environment does not generalize well enough to another network environment.

Therefore, to analyse the problem, the values of the features of the model are calculated. The most important features will be used to compare data from two datasets. The Kolmogorov–Smirnov plots and statistics [87] assume that all features come from different distributions in both datasets, which poses a problem for the estimator because it assumes that the training, test, and real data come from the same distribution. Hence, the Kolmogorov–Smirnov statistic was performed for all features of the datasets. After removing the no variance feature, there are only two features coming from the same distribution in both fwd_urg_flags and cwe_flag_count datasets, both of which are not good predictors. This shows that data from different network environments is distributed differently.

In order to create an estimator that summarizes data well from different network environments, the estimator is created using the combined CIC-IDS-2017 and CIC-IDS-2018 datasets. Both datasets contain attack classes with a small number of cases. To get a higher detection rate for these attacks, Synthetic Minority Oversampling is used to increase the occurrences of these classes to 100,000. For a combined estimation, the gradient boosting model is trained using grid search to find the best set of hyperparameters.

It should be noted that the combined estimator shows promising performance on the test dataset with a high recall of 0.99, a precision of 0.99, and an attack detection rate (recall class 1) of 0.96 (see Figure 9 for combined estimator results).

The following list (see Figure 10) shows the feature importances of the combined estimator calculated with permutation importance.

According to the SHAP analysis [88], the following features have the greatest impact on the model assumptions:init_fwd_win_byts: Number of bytes sent in the initial window in the forward direction.fwd_pkt_len_mean: Mean size of packet in forward direction.protocol: Protocol.init_bwd_win_byts: Number of bytes sent in the initial window in the backward direction.fwd_seg_size_min: Minimum segment size observed in the forward direction.

While the performance of the combined estimator is convincing, it can be assumed that the estimator will not generalize well across different network environments due to observed differences in distributions. The statistical characteristics recorded in the individual datasets appear to be highly dependent on the network topology and the configurations of the host and client machines on the network.

To solve this problem, the following suggestions are offered:The estimator should train on more diverse data coming from different network environments.The evaluator must be trained with data obtained from the network environment in which it will be deployed.

The second option seems to be more promising since it is very difficult to obtain high-quality datasets on real network attacks. The fact that data must be collected in the target environment can be alleviated by collecting only secure network traffic and using an anomaly detection approach to detect network attacks.

### 3.5. Experiment—2

In the second part of the experiment, the authors used an unsupervised learning approach to create a binary classifier based on the ideas of representation learning and anomaly detection. The idea was that several deep learning models were trained on benign data from the CIC-IDS-2018 dataset in order to learn the meaningful representation of these benign data. With this approach, there was a chance to create a model capable of classifying network traffic as safe or malicious, based on the notion of similarity or dissimilarity of the traffic to the data on which the model was trained. The rationale for using unsupervised learning is that useful data is usually easier to obtain and therefore can be provided in larger volumes than malicious data. For this, the authors decided to apply the autoencoder architecture (neural network).

It is known that an autoencoder learns to reconstruct given inputs by initially encoding the input features as dense representations and then decoding the dense representations to reconstruct the original input. Using this approach, the model must learn an income identification function.

As mentioned above, given the use case of network traffic classification, the model is only trained on good data. A secure and malicious data validation set is used to determine a decision boundary based on both types of traffic and the reconstruction errors.

On inference, the sample is fed into the autoencoder, the reconstruction error is measured, and then the sample is classified as malicious if the reconstruction error exceeds a predetermined decision boundary.

For all this, three variants of the autoencoder architecture are taken into account (Undercomplete, Stacked, Denoising).

Summarising the results of the experiments using anomaly detection, it can be seen that the performance of the resulting model is insufficient for real use and significantly worse than the performance of machine learning models created in previous experiments (see Figure 11 to compare performance results of autoencoders).

Moreover, the predictions of this estimator are very sensitive to the chosen value of the decision boundary, which can be reliably determined only if there is a sufficient amount of malicious data. This circumstance somewhat reduces the usefulness of this approach, since the biggest advantage of this method is the assumption that only secure data is needed to create an evaluator, and the collection of malicious data is not required or strictly limited.

The demonstrated approach can be useful in situations where malicious training data is only available in small amounts or is not available at all. If there is no malicious training data, the choice of the decision boundary can be made by determining a reasonable confidence interval taking into account the distribution of safe samples and adjusting the boundary when new data arrives.

### 3.6. Experiment—3

In this experiment, the authors used a supervised learning approach to create a binary classifier capable of distinguishing between safe and malicious network traffic. Several deep neural network models were selected using network traffic data taken from the CIC-IDS-2018 dataset and their respective characteristics were evaluated.

The values of the dataset target variable are grouped into two classes: benign and attacking, while the attack class includes all types of malicious network traffic. Since the dataset is highly unbalanced and contains 83% safe and only 27% malicious samples, this class imbalance is taken into account during training.

In the first part, a simple deep network was trained using two different approaches. The first approach does not take class imbalance into account, while the second approach uses class weights to weight the underrepresented sample loss more heavily during training. Comparing the results of both training runs, one of the two approaches was chosen for further research. The second part was to find the optimal model architecture and configuration parameters for the classifier by optimizing the hyperparameters using the Hyperopt library.

The performance of the static model without class weights is very stable with a PR score of 0.97718. The classification reports show that the positive class has a much higher (0.992) and a lower recall (0.939). This effect can be caused mainly by numerous negative class monsters in the training set. Consequently, the confusion matrix in Figure 12 reveals a few false positives, but a high number of false negatives.

The misclassification statistics show that most of the false-negative results are due to the infiltration of the attack category, with a misclassification rate of 98%.

The performance of the static weighted class model for PR points is 0.97735 points, which is slightly better than the previous model (Figure 13).

Nonetheless, the classification report shows a different combination of precision and recall for the positive class, with a precision of 0.965 and a recall of 0.954. The confusion matrix reveals an almost equal number of false positives (9375 samples) and false negatives (12,649 samples). Although the misclassification rate of infiltration was reduced to 77%, it still is very high. By using class weights during training, it is possible to reduce precision by about 2.7%, but increase recall by 1.6%. As a result, the number of false negatives decreases, and the number of false positives increases by an acceptable amount (Table 3).

Although the second approach does not look so appealing, the authors had to choose it because, at the beginning of the third experiment, the goal was to detect as many attacks as possible by using class weights during training. Additionally, both models suffer from a high level of misclassification against the penetration attack category, likely due to the fact that benign traffic and penetrating traffic are very similar.

In this section, model training with hyperparameter optimization was applied. The Hyperopt library [89] helped to train and evaluate different model architectures in combination with different hyperparameter configurations. For this purpose, it was necessary to decide on a training method that would accept model parameters from the Hyperopt library, dynamically create a model, perform training, and return minimal validation losses.

The best model has the following parameter configuration (Figure 14):5 layers300 units per layerA dropout rate of 0.22Elu activation functionAdam optimizerA learning rate multiplier of 0.61 effectively reduces the default learning rate of 0.001 to 0.00061.

The authors also wanted to perform a second optimization step using the optimal parameter values found in the first round of hyperparameter search. This would allow others to further explore the space of optimal parameters.

For this round, a variable batch size was used, and 20 trials were run. However, the second round of hyperparameter search did not give a better result. In the second optimization, the model loss was fixed at 0.1306, compared to the 0.1302 loss obtained from the best model of the first round. On the other hand, the difference in losses was negligible, which suggests that this is a fairly good configuration for the model.

In the last step of this section, the optimal parameter values and model configuration were used, which had been determined during the first round of hyperparameter searches. The model was trained using the optimal parameters for 200 epochs, with the objective of obtaining better performance. Inspecting the learning curves, it can be seen that the model does not significantly overfit the training set (Figure 15).

After training, a model was found with a slightly lower loss of 0.1298 and a better PR score of 0.97816 compared with the best model found when searching for hyperparameters with a loss of 0.1302 and a PR score of 0.97793 (see Figure 16).

To summarise this experiment, two approaches to train a deep neural network for the task of binary classification of network traffic were investigated. It is important that the training was carried out whilst taking into account the class imbalance in order to minimize the number of false negative results.

In addition, we performed hyper parametric search and optimization to study the optimal network architecture and configuration of parameters, which resulted in a model with impressive performance.

Despite the fact that the model seems to work well with most types of network attacks, it cannot correctly identify penetration attacks, mistakenly classifying 77% of all penetration traffic as secure traffic. This can be explained by the similarity of the statistics of characteristics observed in these two types of network traffic.

As a result, it is necessary to explore other approaches for the reliable detection of penetration attacks.

### 3.7. Results

Ultimately, in order to compare which algorithm is the best to use for intrusion detection systems, a comparative analysis of the best three algorithms was carried out:Random forest classifier using scikit-learnGradient Boosted Tree Classifier using CatBoost libraryDeep neural network using Keras and Tensorflow.

Initially, all models should be in the same positions, therefore the search and optimization of hyperparameters for the Random Forest and Gradient Boosted Tree algorithms were carried out, since in previous experiments the default parameters of both algorithms were used for training. As in previous experiments, the Hyperopt library was used to search by hyperparameters. The data used to train and compare models is from the CIC-IDS-2018 dataset.

After training and fine-tuning, the authors compared the models based on their respective characteristics in the validation set and selected the model with the best performance. This model was subsequently evaluated on a test set to obtain an unbiased estimate of the model’s performance (Figure 17).

The Random Forest and Gradient Boosted Tree models outperformed the neural network, achieving slightly higher PR, precision, and recall (Table 4 Comparison of models based on metrics).

Overall, the Gradient Boosted Tree model offered the best performance, with the highest PR and recall rate compared with the positive class, with only slightly lower accuracy than the random forest model. Therefore, this model also returned the fewest false negatives.

When the most efficient model algorithm was found, it had to be tested and evaluated objectively. As a result, if the performance on the test set was very similar to the performance on the first set, then the data has the same statistics as the test set and it can be assumed that this model generalizes unseen data well (see Figure 18 Performance on Test Set).

In our model’s comparison experiment, the model was built using gradient boosting and tested on the test dataset and was found to perform well on the test data as well. The estimator shows a recall and accuracy of 0.98, which is acceptable for real use. In this study, minority attack classes are often misclassified. SMOTE can be applied to the training dataset for these classes to improve future performance. Additional tests can be performed on new data from different network environments to ensure that the estimator performs the same as it did on the test dataset.

## 4. Discussion and Research Challenges

With the rapid growth of the Internet of Things and the development of fifth generation mobile networks, there are great opportunities for further network cybersecurity research. The approach outlined in this research paper is just one of a few topics that address the security of the entire network. In addition, in order for an intrusion detection system to show good results, it is necessary to conduct further checks and tests on big data and from different gadgets/devices.

In general, during the study, it became clear that the machine learning approach allowed us to automate the process. This topic can be developed further, or an IDS can be created for the 5G core network using various machine learning or deep learning algorithms. It also generated ideas such as:The creation of a new dataset that will collect network traffic. This is very relevant, as some datasets have lost their novelty. After this set, one can research it and build new models;The implementation of a real-time traffic monitoring system;The application of a Machine Learning Approach to the Internet of Things;The protection of the machine learning system from potential hackers. For example, they can access 5G databases using vulnerabilities. Once they have access, they can use machine learning techniques to obtain sensitive information;The use of semi-supervised machine learning models—in reality, not many datasets have labeled data;The investigation of the possibility of detecting enemy attacks that lead to misleading predictions of unknown attack types;The observation of communication and security standards in mobile applications.

## 5. Conclusions

In this research paper, a comparative analysis of the application of machine learning and deep learning for a network intrusion detection system in a 5G network was carried out. However, prior to building the model, an overview was made of popular and recent datasets for tracking normal and malicious network traffic. As a result, two datasets were selected (CICIDS2017 and CSE-CIC-IDS2018).

The following models were trained and evaluated: Logistic Regression, Random Forest, Gradient Boosting, Autoencoder, and DNN with hyperparameter search. The whole process was divided into three parts and only one ML or DL algorithm was chosen in each experiment, meaning that in the end they could compete with each other. In the first part, the Gradient Boost algorithm performed well, showing a recall of 0.99 and a precision of 0.99. In the second part, when the autoencoder models were built, it was revealed that the performance of the resulting model was insufficient for real use and was much worse than the performance of the machine learning models. In the third part, two approaches for training a deep neural network for the task of binary classification of network traffic were explored.

In the end, according to the best results based on metrics, Gradient Boost came out as the best algorithm, showing on the test set 99.3% for a secure dataset and 96.4% for attacks.

An intrusion detection system (IDS) installed in a network examines network activity to learn about potential threats and vulnerabilities that could harm the system and the network environment. It is anticipated that IDS will be built into the core of the 5G network as a new network feature. This methodology is also proposed for subsequent generations, 6G and 7G, as the architecture of this generation involves integrating advanced features into the existing 5G technology to perform tasks at the individual and group levels.

## Figures and Tables

**Figure 1 sensors-22-09957-f001:**
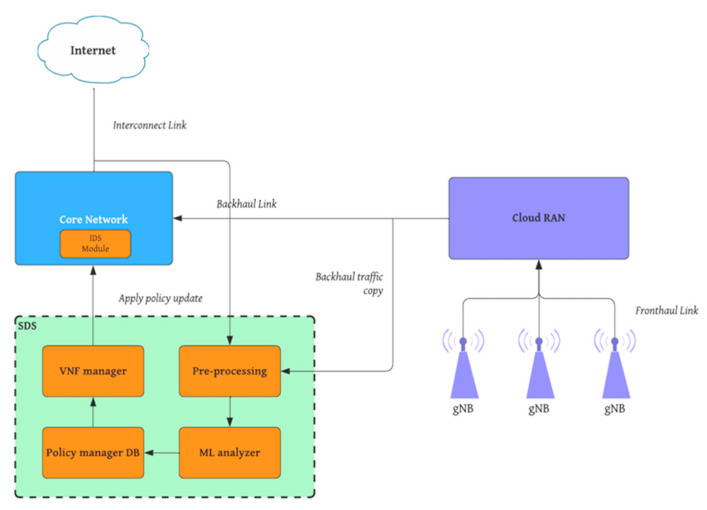
5G network with IDS module system.

**Figure 2 sensors-22-09957-f002:**
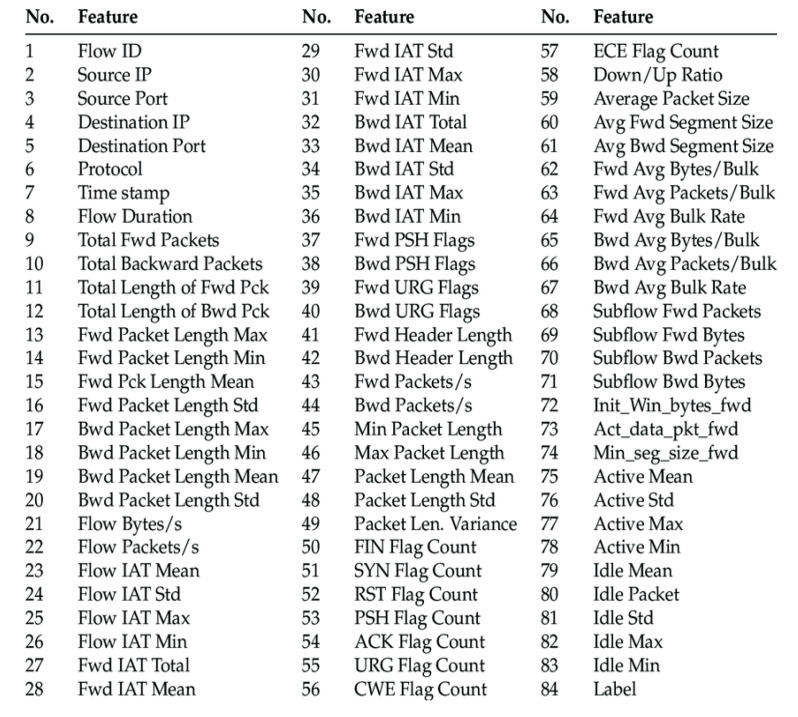
CICIDS2017 and CSE-CIC-IDS2018 network traffic features [67].

**Figure 3 sensors-22-09957-f003:**
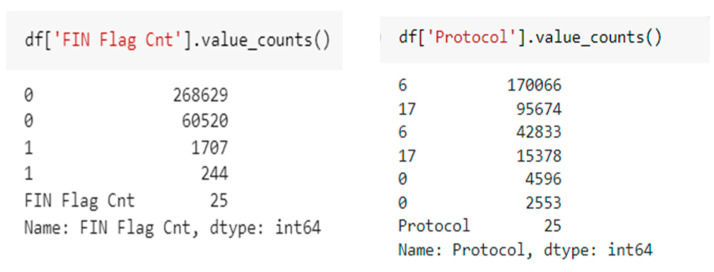
info() method.

**Figure 4 sensors-22-09957-f004:**
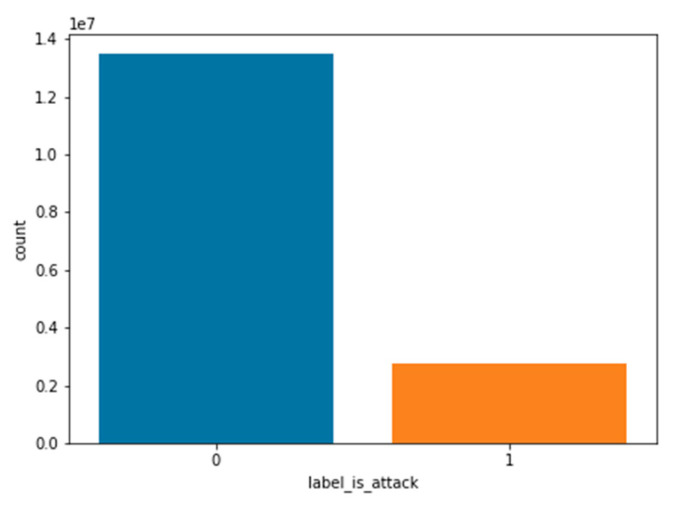
Number of benign and malicious flows.

**Figure 5 sensors-22-09957-f005:**
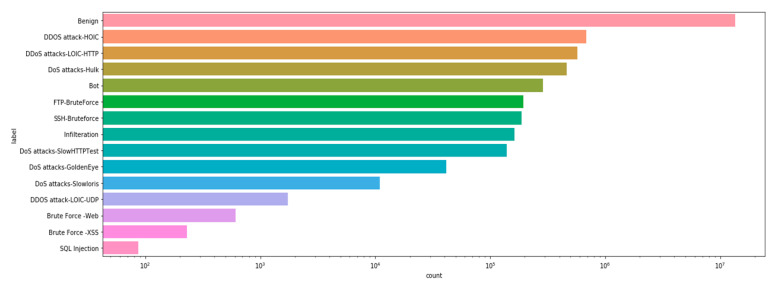
Number of flows/attack type.

**Figure 6 sensors-22-09957-f006:**
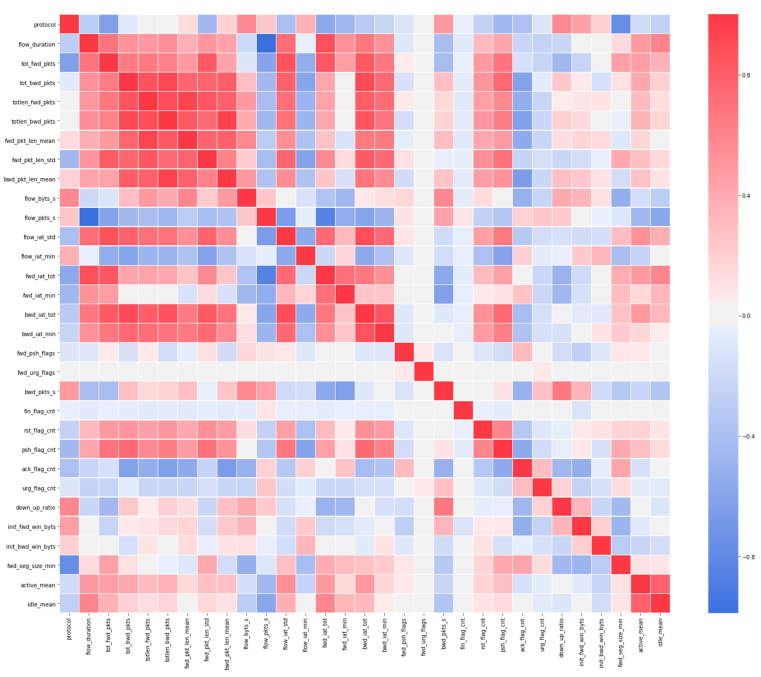
Correlation heatmap after the removal of highly correlated features.

**Figure 7 sensors-22-09957-f007:**
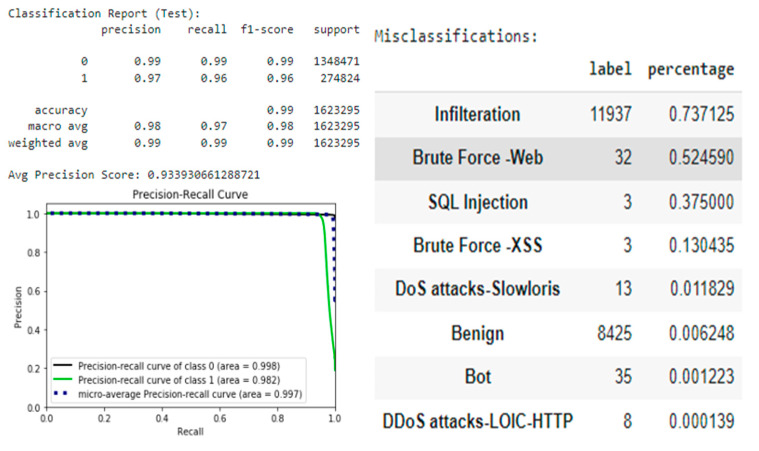
Gradient Boost performance on test dataset.

**Figure 8 sensors-22-09957-f008:**
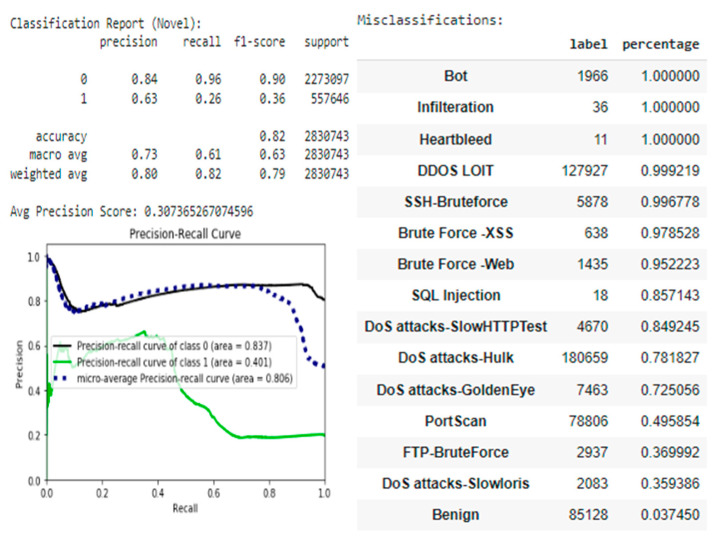
Testing on novel data.

**Figure 9 sensors-22-09957-f009:**
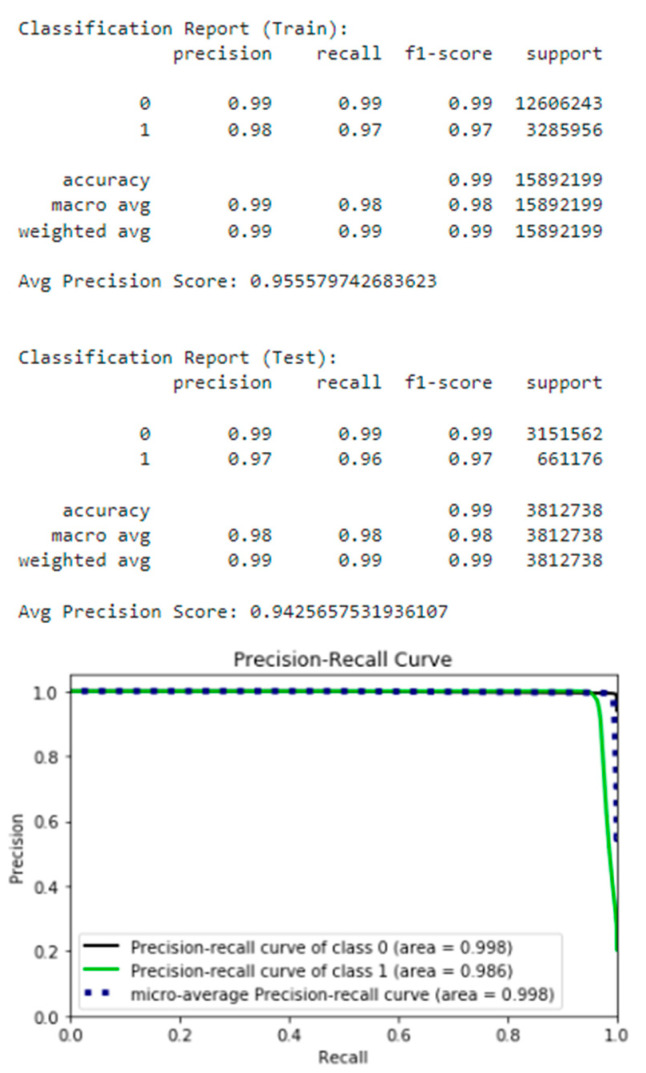
Combined estimator.

**Figure 10 sensors-22-09957-f010:**
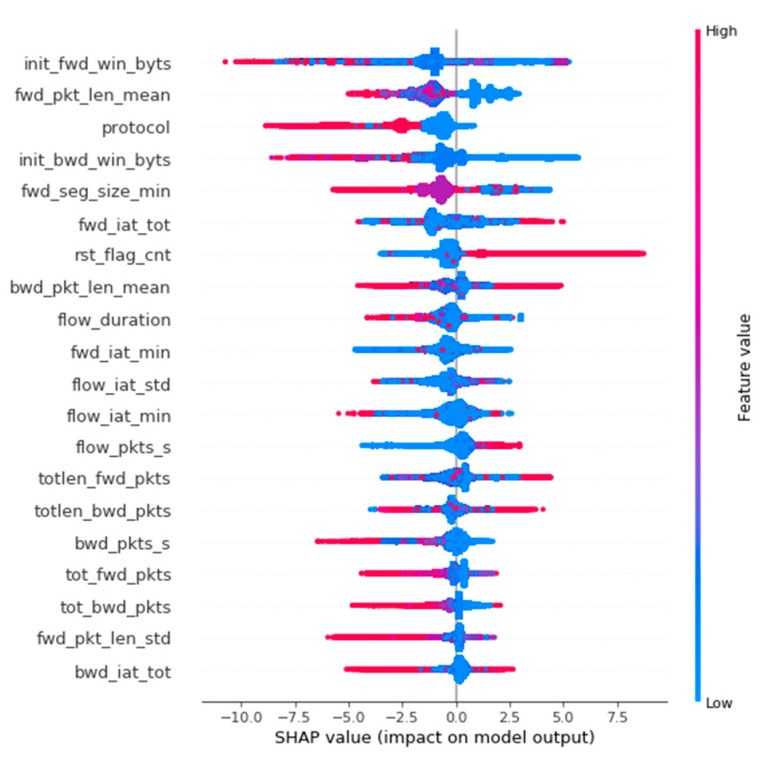
Feature importance.

**Figure 11 sensors-22-09957-f011:**
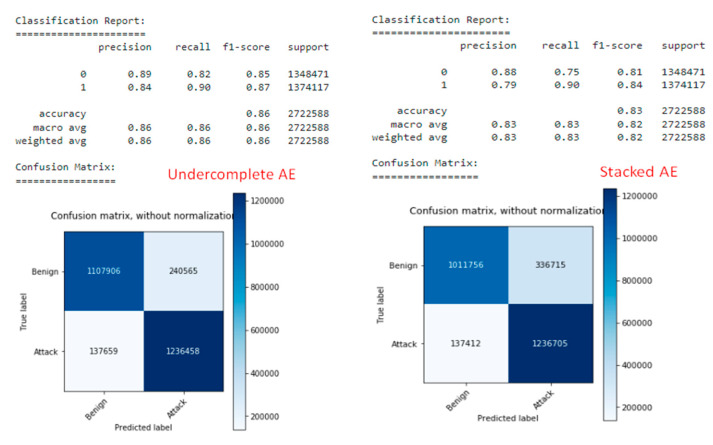
Performance of Undercomplete, Stacked, Denoising Autoencoders.

**Figure 12 sensors-22-09957-f012:**
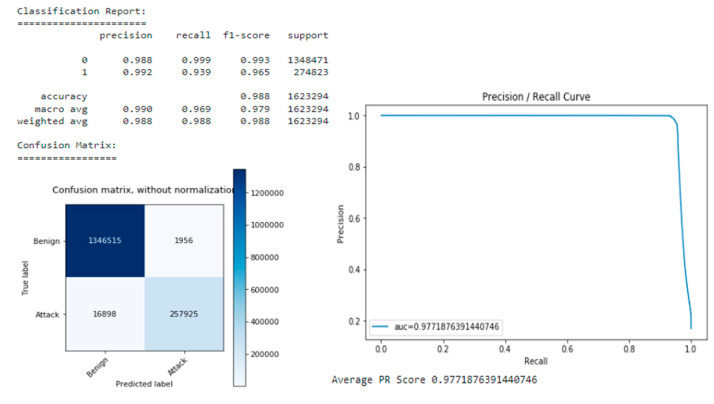
Static Model without class weights.

**Figure 13 sensors-22-09957-f013:**
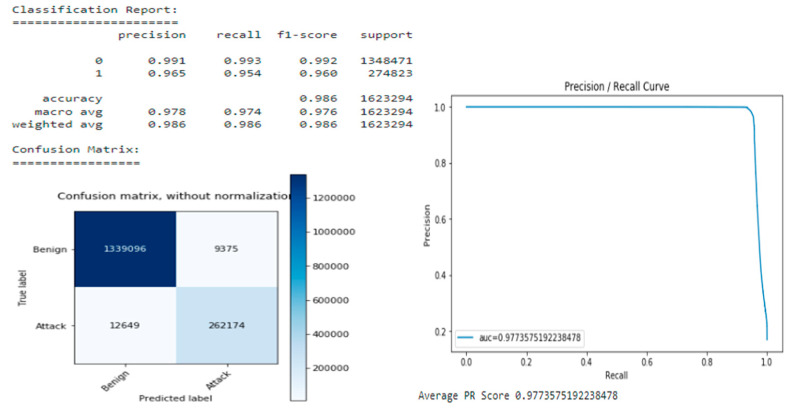
Static Model using class weights.

**Figure 14 sensors-22-09957-f014:**
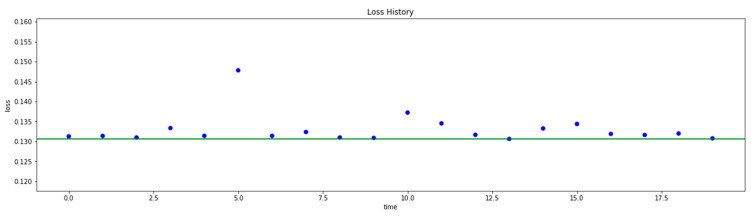
DNN Model Training with Hyperparameter Optimization.

**Figure 15 sensors-22-09957-f015:**
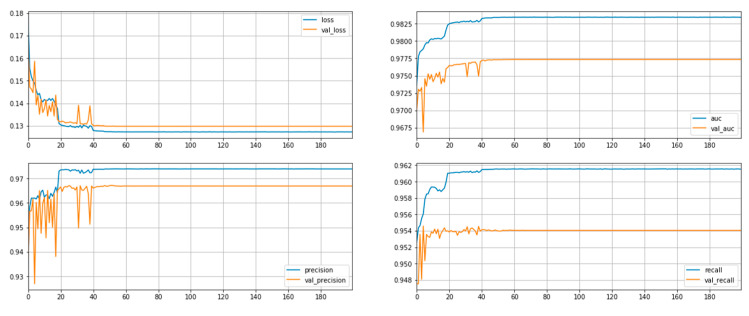
Learning curves of optimized model.

**Figure 16 sensors-22-09957-f016:**
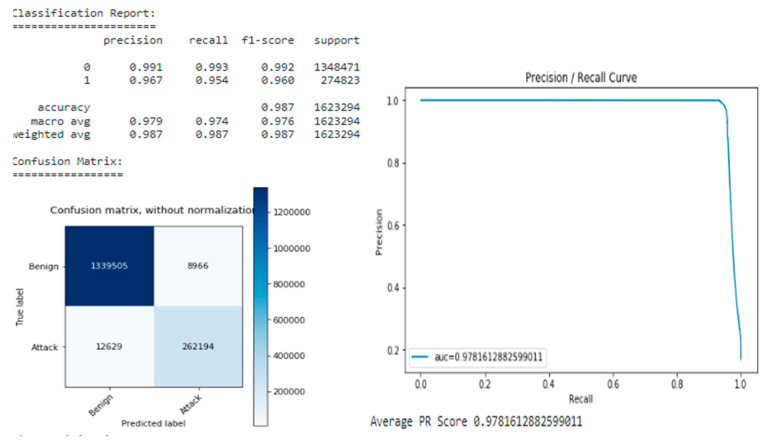
Model with Optimal Parameters.

**Figure 17 sensors-22-09957-f017:**
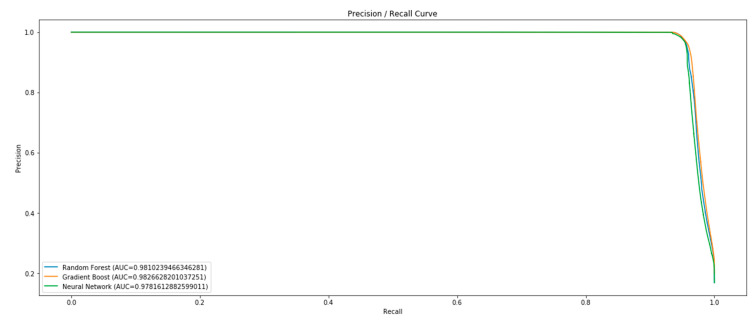
Precision/Recall curves of the different models.

**Figure 18 sensors-22-09957-f018:**
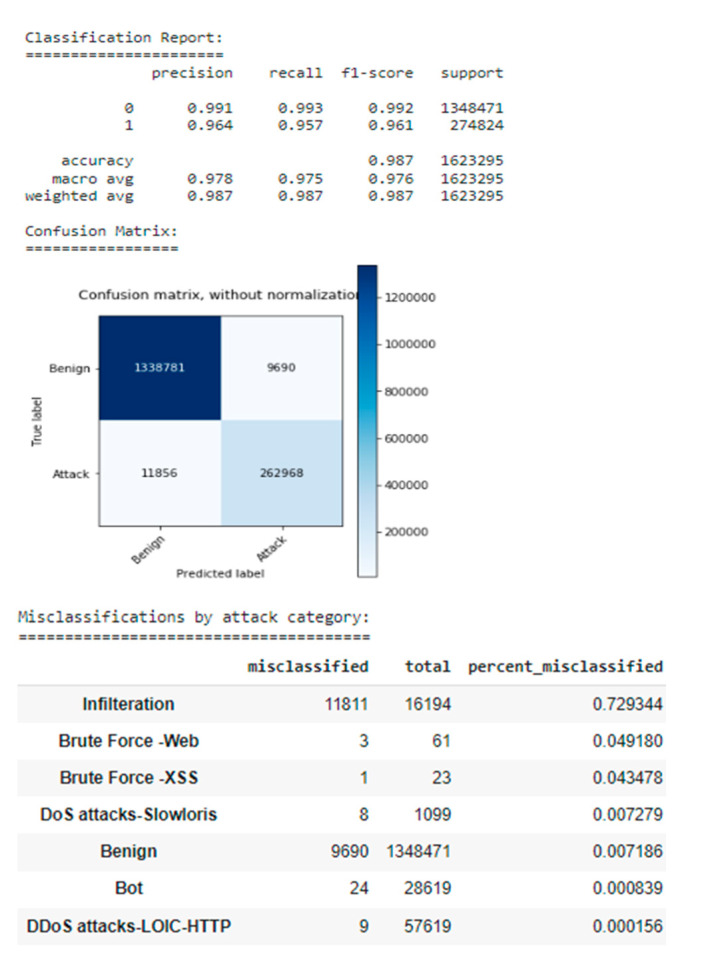
Performance on Test Set.

**Table 1 sensors-22-09957-t001:** Results for training dataset.

No. of Selected Features: 31		Precision	Recall	F1-Score	Support	Precision	Recall	F1-Score	Support
Technique	Accuracy	Class 0				Class 1			
Logistic Regression	0.88	0.90	0.95	0.93	10,787,766	0.68	0.49	0.57	2,198,588
Random Forest	0.99	1.00	0.99	0.99	10,787,766	0.96	0.98	0.97	2,198,588
Gradient Boosting	0.99	0.99	0.99	0.99	107,877,66	0.97	0.96	0.96	2,198,588

**Table 2 sensors-22-09957-t002:** Model selection.

Model	Precision Benign (0)	Recall Benign (0)	F1-Score	Avg Precision	Recall Attack (1)	Precision Attack (1)
Logistic Regression	0.86	0.88	0.87	0.422	0.49	0.68
Random Forest	0.99	0.99	0.99	0.925	0.95	0.96
Gradient Boosting	0.99	0.99	0.99	0.933	0.96	0.97

**Table 3 sensors-22-09957-t003:** DNN models comparison.

Model	PR Score	Precision Positive	Recall Positive	False-Positives	False-Negatives
**Static model (no class-weights)**	0.97718	**0.992**	0.939	**1956**	16,898
**Static model (class-weights)**	0.97735	0.965	**0.954**	9375	12,649
**Optimized model (class-weights)**	**0.97816**	0.967	**0.954**	8966	**12,629**

**Table 4 sensors-22-09957-t004:** Comparison of models based on metrics.

Model	PR Score	Precision Positive	Recall Positive	False-Positives	False-Negatives
**Random Forest**	0.98102	**0.967**	0.955	**8820**	12,322
**Gradient Boosted Trees**	**0.98266**	0.964	**0.957**	9784	**11,748**
**Deep Neural Network**	0.97816	**0.967**	0.954	8966	12,629

## Data Availability

Publicly available datasets were analyzed in this study. This data can be found here: https://www.unb.ca/cic/datasets/ids-2017.html, https://www.unb.ca/cic/datasets/ids-2018.html accessed on 14 December 2020.

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
