# Peer review of "Research of Machine Learning Algorithms for the Development of Intrusion Detection Systems in 5G Mobile Networks and Beyond"

_sensors, 2022, doi:10.3390/s22249957_

Reviewer 1 Report
This is an interesting and actual work that proposes to implement an intrusion detection by applying the machine learning approach to the 5G architecture. Generally, this material is divided into two parts: the first part is the analysis of datasets for intrusion detection in networks, and the second one is the application of machine learning algorithms to secure the 5G network. Gradient boosting has been experimentally approved to be the top technique, performing 96% accuracy for the attack detection. Contribution of this work is the structured approach to select a ML method for intrusion detection in the 5G networks. Merit of this work is the experimental outcomes of the tested ML methods. While the contribution of the reviewed work is acknowledged, there are a few drawbacks that make this material weaker:
1) This paper should be placed next correctly with other related works. There is no comparison of the authors’ work presented in this manuscript against near works in the same area of research. Section 1.1. is too narrow for the wide area of the related works in the 5G intrusion detection. No work placement hardens the authors’ contribution novelty be understood. The authors are recommended to provide a separate section “The related works” or in the mandatory “Discussion” section where to compare their approach with the proposed by other scholars and researchers.
2) The paper announces the ML techniques for intrusion detection in “…5G mobile networks and beyond”. Unfortunately, there is no 5G-relative specifics stated in their contribution. As well, there is no “...and beyond” (e.g., 6G or 7G) specifics in their research. There are no specific attacks typical for 5G networks and no typical for “5G and beyond” datasets were examined while ML testing and comparing. Therefore, the reader can see that the title and aim of work do not correspond to its content and result. Authors should specify their approach applied exactly with attention to 5G/6G/7G networks as well as specific intrusions relative to 5G/6G/7G infrastructures.
3) CICIDS2017 and CSE-CIC-IDS-2018 datasets have been selected in the work for ML methods testing. But they are the ordinary datasets of regular computer networks that do not match the 5G/6G/7G’s security threat model. Authors should specify the originality and novelty of their ML method selection applied exactly to 5G/6G/7G traffic. Also, while selecting the dataset for testing the authors should specify why they do select that dataset and not others.
4) Logistic Regression (LR), Gradient Boosting, Random Forest (RF), Autoencoder and Deep Neural Network (DNN) were selected for analysis. And what about other ML-relative classifiers? One issue typical for 5G/6G/7G is the ‘big data explosion’. And in genuine not every ML method can correctly and freely process big data input. There is no information found in the manuscript concerning this challenge. Authors should analyze this problem and select the ML mechanism concerning the real 5G security challenge.
5) Little presentation errors in the material:
a. datasets are disbalanced. As disbalanced dataset results in high error rate, how are they applied with low error rate?
b. Section 2.1 presents well-known information and can be replaces by appropriate book references.
c. Figure 2 and Figure 3 plot the same info (for different datasets, but same essence) and can be merged.
d. Figure 4 can be presented textually to save the place for the above mentioned issues.
e. Figure 7 and Figure 8 present are too practical; they plot not so useful info for the goal of this contribution.
f. Title of Table 1 and table itself are on different pages. As well as Figure 22, title and drawing are divided.
g. Strings 178-184 and 194-200 are duplicated.
Therefore, this manuscript is recommended to be accepted, but after the revision.
Author Response
Dear Academic Editor,
Thank you for giving me the opportunity to submit a revised draft of our manuscript titled “Research of machine learning algorithms for the development of intrusion detection systems in 5G mobile networks and beyond” to Sensors. We are grateful to you for your insightful comments on our paper. We have been able to incorporate changes to reflect most of the suggestions provided by you. We have highlighted the changes within the manuscript.
Here is a point-by-point response to your comments and concerns.
This is an interesting and actual work that proposes to implement an intrusion detection by applying the machine learning approach to the 5G architecture. Generally, this material is divided into two parts: the first part is the analysis of datasets for intrusion detection in networks, and the second one is the application of machine learning algorithms to secure the 5G network. Gradient boosting has been experimentally approved to be the top technique, performing 96% accuracy for the attack detection. Contribution of this work is the structured approach to select a ML method for intrusion detection in the 5G networks. Merit of this work is the experimental outcomes of the tested ML methods. While the contribution of the reviewed work is acknowledged, there are a few drawbacks that make this material weaker:
1. This paper should be placed next correctly with other related works. There is no comparison of the authors’ work presented in this manuscript against near works in the same area of research. Section 1.1. is too narrow for the wide area of the related works in the 5G intrusion detection. No work placement hardens the authors’ contribution novelty be understood. The authors are recommended to provide a separate section “The related works” or in the mandatory “Discussion” section where to compare their approach with the proposed by other scholars and researchers.
Response 1: Thank you for pointing this out. We agree with this comment. Therefore, we have added new part 1.2 Related works, where we have made an overview of works related to our study, also for the sake of clarity we have added information in 1. Introduction and 1.1 Background analysis. All corrections are shown in red.
2. The paper announces the ML techniques for intrusion detection in “…5G mobile networks and beyond”. Unfortunately, there is no 5G-relative specifics stated in their contribution. As well, there is no “...and beyond” (e.g., 6G or 7G) specifics in their research. There are no specific attacks typical for 5G networks and no typical for “5G and beyond” datasets were examined while ML testing and comparing. Therefore, the reader can see that the title and aim of work do not correspond to its content and result. Authors should specify their approach applied exactly with attention to 5G/6G/7G networks as well as specific intrusions relative to 5G/6G/7G infrastructures.
Response 2: 5G inherits vulnerabilities from earlier technologies and similarly 5G vulnerabilities can be inherited by 6G networks. More details added in 1.1 Background analysis.
3. CICIDS2017 and CSE-CIC-IDS-2018 datasets have been selected in the work for ML methods testing. But they are the ordinary datasets of regular computer networks that do not match the 5G/6G/7G’s security threat model. Authors should specify the originality and novelty of their ML method selection applied exactly to 5G/6G/7G traffic. Also, while selecting the dataset for testing the authors should specify why they do select that dataset and not others.
Response 3: Agree. We have, accordingly, changed part 2.Proposed Methodology and Model classifiers and add 291-306 strings to emphasize this point.
4. Logistic Regression (LR), Gradient Boosting, Random Forest (RF), Autoencoder and Deep Neural Network (DNN) were selected for analysis. And what about other ML-relative classifiers? One issue typical for 5G/6G/7G is the ‘big data explosion’. And in genuine not every ML method can correctly and freely process big data input. There is no information found in the manuscript concerning this challenge. Authors should analyze this problem and select the ML mechanism concerning the real 5G security challenge.
Response 4: Agree. We have, accordingly, changed part Building ML model prototype 430-445 strings to emphasize this point.
5. Little presentation errors in the material:
a. datasets are disbalanced. As disbalanced dataset results in high error rate, how are they applied with low error rate?
Response: In the first part, the model was trained using two different approaches. The first approach does not consider the class imbalance, while the second approach uses the class weights to weigh more heavily the underrepresented sample loss during training.
b. Section 2.1 presents well-known information and can be replaces by appropriate book references.
Response: Agree. We have, accordingly, revised and shortened the information, also inserting references to literature.
c. Figure 2 and Figure 3 plot the same info (for different datasets, but same essence) and can be merged.
Response: Corrected.
d. Figure 4 can be presented textually to save the place for the above mentioned issues.
Response: Corrected.
e. Figure 7 and Figure 8 present are too practical; they plot not so useful info for the goal of this contribution.
Response: We put that data on appendix A.
f. Title of Table 1 and table itself are on different pages. As well as Figure 22, title and drawing are divided.
Response: Corrected.
g. Strings 178-184 and 194-200 are duplicated. Herefore, this manuscript is recommended to be accepted, but after the revision.
Response: Thank you for pointing this out. We have removed duplicate information.
We look forward to hearing from you in due time regarding our submission and to respond to any further questions and comments you may have.
Sincerely,
Azamat Imanbayev.
Reviewer 2 Report
* The manuscript presents an approach to detect intrusion in 5G mobile network and to verify the performance of the proposed approach two datasets have been used namely CICIDS-2017 and CSE-CIC-IDS-2018. How are these two datasets related to 5G mobile network? This is because these datasets are extensively used to verify the approach proposed for IDS.
* It is difficult to identify contributions and novelty of the proposed approach. The manuscript should have a paragraph stating novelty and contributions.
* Related section work needs significant improvements as many recent useful references are missing. The following may be useful for the same.
->A review of the advancement in intrusion detection datasets
-> A review on machine learning and deep learning perspectives of IDS for IoT: recent updates, security issues, and challenges
-> Attack classification using feature selection techniques: a comparative study
-> Role of swarm and evolutionary algorithms for intrusion detection system: A survey
-> A survey on intrusion detection system: feature selection, model, performance measures, application perspective, challenges, and future research directions
-> Application domains, evaluation data sets, and research challenges of IoT: A Systematic Review
-> Intrusion detection using deep neural network with antirectifier layer
-> Analyzing fusion of regularization techniques in the deep learning‐based intrusion detection system
-> Fusion of statistical importance for feature selection in Deep Neural Network-based Intrusion Detection System
->A survey on different network intrusion detection systems and countermeasure
* Code should be provided as a separate file rather than providing screenshot of the code.
* Result should be presented in meaningful way along with statistical significance of the same.
Author Response
Dear Academic Editor,
Thank you for giving me the opportunity to submit a revised draft of our manuscript titled “Research of machine learning algorithms for the development of intrusion detection systems in 5G mobile networks and beyond” to Sensors. We are grateful to you for your insightful comments on our paper. We have been able to incorporate changes to reflect most of the suggestions provided by you. We have highlighted the changes within the manuscript.
Here is a point-by-point response to your comments and concerns.
1. The manuscript presents an approach to detect intrusion in 5G mobile network and to verify the performance of the proposed approach two datasets have been used namely CICIDS-2017 and CSE-CIC-IDS-2018. How are these two datasets related to 5G mobile network? This is because these datasets are extensively used to verify the approach proposed for IDS.
Response 1: Agree. We have, accordingly, changed part 2.Proposed Methodology and Model classifiers and add 291-306 strings to emphasize this point.
2. It is difficult to identify contributions and novelty of the proposed approach. The manuscript should have a paragraph stating novelty and contributions.
Response 2: We propose a DNN-based intrusion detection and classification system, considering statistical measures to evaluate the performance of models. We applied statistical importance fusion using statistical measures to obtain association and determine feature significance for feature selection. For implementation purposes, two public data sets are used, such as CICIDS2017 and CSE-CIC-IDS2018. We added at the end of 1.2 Related Works our purpose.
3. Related section work needs significant improvements as many recent useful references are missing. The following may be useful for the same.
Response 3: Thank you for pointing this out. We agree with this comment. Therefore, we have added new part 1.2 Related works, where we have made an overview of works related to our study, also for the sake of clarity we have added information in 1. Introduction and 1.1 Background analysis. All corrections are shown in red.
4. Code should be provided as a separate file rather than providing screenshot of the code.
Response 4: We put that data on the appendix B.
5. Result should be presented in meaningful way along with statistical significance of the same.
Response 5: In our model's comparison experiment, the model is built using gradient boosting and tested on the test dataset and found to perform well on the test data as well. The estimator shows a recall and accuracy of 0.98, which is acceptable for real use. In this study, minority attack classes are often misclassified. SMOTE can be applied to the training dataset for these classes to improve future performance. Additional tests can be performed on new data from different network environments to ensure that the estimator performs the same as it did on the test dataset.
We look forward to hearing from you in due time regarding our submission and to respond to any further questions and comments you may have.
Sincerely,
Azamat Imanbayev.
Round 2
Reviewer 1 Report
The revised version of the manuscript has significant improvements in details: the related works have been observed, experimental data and results have been revised and double-checked, all reviewers question and comments have been worked out and responded. Therefore, the manuscript is well improved comparing it with its firstly submitted version, and the revised version can be recommended to be published.
Author Response
Thank you for giving me the opportunity to submit a revised draft of our manuscript titled “Research of machine learning algorithms for the development of intrusion detection systems in 5G mobile networks and beyond” to Sensors.
We made a few changes to the article because there were comments from the reviewer. We have highlighted the changes within the manuscript.
Reviewer's comment 1: The manuscript is improved compared to its previous version. However, results presented in Figures 7 to 10 and other such figures can be represented in a single table for a better comparison.
Response: We agree, and we removed Figures 7-10 and created a single table where we reflected all the data in the figures above (String 487). And describe in 461-479 strings to emphasize this point. All corrections are shown in red.
Reviewer's comment 2: The manuscript should be proofread by a native speaker.
Response: We agree with the reviewer’s assessment. Accordingly, throughout the manuscript, we have revised after native speaker has proofread and corrected grammar.
We would like to thank you for taking the necessary time and effort to review the manuscript. We sincerely appreciate all your valuable comments and suggestions, which helped us in improving the quality of the manuscript.
Reviewer 2 Report
The manuscript is improved compared to its previous version. However, results presented in Figures 7 to 10 and other such figures can be represented in a single table for a better comparison.
The manuscript should be proofread by a native speaker.
Author Response
Dear Academic Editor,
Thank you for giving me the opportunity to submit a revised draft of our manuscript titled “Research of machine learning algorithms for the development of intrusion detection systems in 5G mobile networks and beyond” to Sensors. We are grateful to you for your insightful comments on our paper. We have been able to incorporate changes to reflect most of the suggestions provided by you. We have highlighted the changes within the manuscript.
Here is a point-by-point response to your comments and concerns.
The manuscript is improved compared to its previous version. However, results presented in Figures 7 to 10 and other such figures can be represented in a single table for a better comparison.
Response: Thank you for pointing this out. We agree, and we removed Figures 7-10 and created a single table where we reflected all the data in the figures above (String 487). And describe in 461-479 strings to emphasize this point. All corrections are shown in red.
The manuscript should be proofread by a native speaker.
Response: We agree with the reviewer’s assessment. Accordingly, throughout the manuscript, we have revised after native speaker has proofread and corrected grammar.
We would like to take this opportunity to thank you for the effort and expertise that you contributed towards reviewing the article, without which it would be impossible to maintain the high standards of peer-reviewed journals.